# Macro and Microelements in Leaves of 'Meredith' Peach Cultivar Supplied with Biochar, Organic and Beneficial Biofertilizer Combinations

**Mateusz Frąc [1,\*][ID], Lidia Sas-Paszt [1][ID] and Mirosław Sitarek [2][ID]**

1 Department of Microbiology and Rhizosphere, The National Institute of Horticultural Research, Konstytucji 3 Maja 1/3, 96-100 Skierniewice, Poland
2 Department of Cultivars Testing, Nursery and Genetic Resources, The National Institute of Horticultural Research, Konstytucji 3 Maja 1/3, 96-100 Skierniewice, Poland
\* Correspondence: mateusz.frac@inhort.pl

**Abstract:** The content of macro and microelements in the leaves of peach trees treated with biochar, organic fertilization and microorganisms in the field experiment was tested. The experiment was carried out in accordance with the integrated fruit production methods at the NIHR Experimental Orchard in Dąbrowice, from 2015 to 2017. The trees were grafted on *P. persica* Mandżurska root-stock and planted in the spring of 2013. In 2014, the following products were applied around the trees and mixed into the topsoil: biochar at a dose of 1.6 kg/tree (2000 kg/ha); biochar at a dose of 1.6 kg/tree used together with microorganisms—bacteria *Pseudomonas fluorescens*, *Pantoea* and arbuscular mycorrhizal fungi—*Glomus caledonium*, *Glomus intraradices* and *Glomus coronatum*; biochar at a dose of 1.6 kg/tree applied together with Florovit NPK organic fertilizer at a dose of 0.2 kg/tree; Florovit NPK; microorganisms—bacteria *Pseudomonas fluorescens*, *Pantoea* and arbuscular mycorrhizal fungi—*Glomus caledonium*, *Glomus intraradices* and *Glomus coronatum*; Florovit NPK organic fertilizer with the same microorganisms; and an untreated control. The average results showed that, compared to the control, the biochar increased the nitrogen content in the leaves by 6%. All experimental combinations increased the content of P and K in the leaves. The most effective at increasing the content of phosphorus in leaves—by 48%—was Florovit. The greatest increase in potassium was after the use of biochar with Florovit—by 38%. The magnesium content ranged from 0.49 to 0.59 g/100 g DW. The highest content of Mg was found in the leaves after the application of biochar with Florovit, and the lowest after the use of biochar alone. The leaves of the trees fertilized with the Florovit organic fertilizer had the lowest calcium content, while the highest calcium content was found in combination with trees treated with biochar only. The use of biochar alone did not increase the content of boron in the leaves compared to the control. In all other combinations, a higher amount of boron was found. The highest—18% more than the control of this microelement had leaves where biochar and Florovit were used. Trees treated with biochar and microorganisms accumulated the least copper in the leaves, while the highest content of this element was found in the combination where biochar fertilization was applied together with Florovit. Lower iron concentrations in peach leaves were found as a result of applying microorganisms, microorganisms with the organic fertilizer and biochar, relative to organic fertilization and the control combination. Trees where only microorganisms were applied to the soil had the least manganese and zinc in the leaves. The accumulation of manganese in the leaves was most favored by fertilization with biochar together with microorganisms, and the zinc content was the highest after the use of biochar alone. In general, studies have shown that a small dose of biochar alone or biochar together with organic fertilizer is a very effective method of feeding peach trees. More research is needed on the use of microorganisms and methods of their application with various products used in orchard fertilization.

**Keywords:** biochar; peach; organic matter; organic fertilizer; microorganism





## 1. Introduction

The feeding of fruit plants based solely on mineral fertilization, without the use of organic fertilizers, composts or bio-fertilizers, leads to deterioration in fruit quality, soil fatigue and depletion in soil organic matter. As a result, the physical and chemical properties of the soil deteriorate, as does the condition and health of the plants growing in it. A visible symptom of this process is the reduction in yields and their lower quality.

The specificity of horticultural cultivation is a long—sometimes extending over several dozen years—period of plant growth in the same location, without the possibility of crop rotation. This leads to a reduction in soil biodiversity, a decrease in the level of activity of soil macro and microorganisms and, consequently, to a general deterioration in fruit quality and soil fertility.

The physical structure of biochar, the mineral content and its chemical properties positively affect the improvement in bio–physico–chemical processes in the soil, increasing the levels of organic carbon and the biomass of beneficial soil mesofauna and microflora.

The application of biochar to the soil brings benefits in terms of improving soil fertility, increasing plant production and by increasing sequestration of greenhouse gases [1]. This has been confirmed by a study by Genesio et al. [2], who investigated the response of grapevines after applying biochar to the soil. Their four-year study showed increased plant productivity following the use of biochar, compared to the control, but they did not observe a positive effect on the quality of the fruit produced.

Aggangan et al. [3] found in their study that the use of arbuscular mycorrhizal fungi (AMF) and biochar stimulated the yielding of agricultural plants and the circulation of minerals in the soil. The bamboo-derived biochar used had a positive effect on increasing the population size of nitrogen-fixing bacteria (NFB) and phosphate-solubilizing bacteria (PSB). Their research indicates a beneficial effect of the use of biochar on the vegetative growth of cocoa seedlings. The research conducted by Bokszczanin et al. [4] showed a positive effect of the use of arbuscular micrite fungi (AMF) and plant growth-promoting rhizobacteria (PGPR) on increasing the content of nitrogen, phosphorus and potassium in apple leaves. Therefore, their use may have a positive effect on the apple yield and fruit quality. The influence of beneficial microorganisms, including the bacteria Pseudomonas fluorescens, can positively affect the growth and development of plants, but also shows high variability (Jacoby et al. [5] after Haney et al. [6]). More and more research is being conducted towards the introduction of microorganisms into the soil, the task of which is to stimulate the growth of crops and to protect crops against pathogens. Suresh et al. [7] showed a positive effect of the use of Pseudomonas fluorescens on tomato growth seedlings. On the other hand, the combined use of Pseudomonas fluorescens with diacetylphloroglucinol (DAPG) contributed to the reduction of the incidence of bacterial wilt.

The research by Prasad et al. [8] shows that not only the type of raw material from which biochar was produced, but also the size of the fraction applied may have a significant impact on the availability of magnesium (Mg) and calcium (Ca) to plants. The biochars used by these authors had a similar pH (8.84–9.56), while the concentrations of phosphorus (P) and potassium (K) were at very different levels, the differences in phosphorus content being especially significant.

Abideen et al. [9], while examining the effects of biochar and biochar with compost on the growth of *Phragmites karka*, observed that the application of biochar to the soil improved the mineral nutrition of plants and increased leaf gas exchange and leaf firmness, compared to control plants. In the experiment by these authors, the combined use of biochar and compost improved the physiological parameters of plants, such as the intensity of photosynthesis, leaf turgor and vegetative growth. Gas exchange and the level of minerals in the soil after the application of biochar and compost were higher than in the control. Baldi et al. [10], using composts from municipal solid waste, did not observe increased concentrations of heavy metals in the leaves and the fruit harvested from nectarine trees. The compost used did not significantly modify the concentrations of macro and microelements in the leaves. However, the concentration of minerals in the fruit was higher

after using the compost fertilization, compared with the fruit harvested from the control plants. Zhu et al. [11], having studied the effects of mineral and organic fertilization on soil fertility and on the biomass and diversity of soil microorganisms, concluded that the combined use of organic and mineral fertilization gave the best results. On the other hand, mineral fertilization on its own produced worse effects in terms of plant growth and yielding than organic fertilization. In the experiments by Adekiya et al. [12], the combined use of biochar and cow manure improved plant growth, yielding and root length, and changed the concentrations of minerals in radish leaves. The use of biochar and compost increased the efficiency of the uptake of nutrients and water by maize plants, and also had a positive effect on plant growth and soil fertility [13]. The use of biochar is recommended mainly on sandy soils to improve their bio–physico–chemical properties. According to the research by Amin and Eissa [14], the application of biochar in highly varied doses to a sandy soil significantly increased the concentrations of nitrogen (N) and phosphorus (P) in zucchini plants. As the dose of biochar increased, the concentrations of these components in plants also increased. The addition of biochar to the soil at a dose of 5% (*v/v*) improved the germination and development of *Sida hermaphrodita* R. in the early stage of plant growth, but decreased the concentrations of phosphorus (P) and calcium (Ca) in the leaves of the plant [15].

In connection with the global trend of limiting the use of mineral (artificial) fertilizers in agriculture, research is being undertaken on the possibility of reducing their doses or replacing them with other products that do not contaminate the soil and groundwater, without a negative impact on plant nutrition. The application of biochar and beneficial microorganisms to the soil is one of the methods whose effectiveness is being tested in the cultivation of various plant species.

The aim of the study was to determine the effects of biochar, organic fertilization and beneficial microorganisms on the concentrations of macro and microelements in the leaves of 'Meredith' peach trees.

This article presents the results of research on changes in the content of mineral components in leaves in three consecutive years. There is little information in the literature about this type of research in field conditions in an orchard. The obtained results can be used in the development of microbiologically enriched fertilizers and peach fertilization programs that are safe for the natural environment.

## 2. Materials and Methods

'Meredith' peach trees grafted on *P. persica* Mandżurska rootstock, on which the experiment was carried out, were planted in the spring of 2013 at a spacing of $4 \times 2$ m, in the Experimental Orchard of the Research Institute of Horticulture in Dąbrowice, central Poland (51°54′51.5″ N 20°06′29.8″ E). In the spring of the second year after planting the trees, combinations with biochar, organic fertilization and beneficial microorganisms were applied to the soil first time. Peach trees grew on a sandy loam podzolic soil classified as class III b, which had been in the monoculture of fruit crops for many years.

The soil had a humus content of 1.4% and a pH of 6.2. The composition of the biochar used in the study, produced by the company Fluid, is presented in Table 1. The experiment was designed in four replications in a random block layout, with 3 trees per plot (replication). The following experimental combinations (treatments) were included:

1. Biochar, applied at a dose of 1.6 kg/tree (2000 kg/ha), produced from coniferous wood chips by the method of fast pyrolysis (at a temperature of 280 °C for 5 min), containing 80% organic matter and 20% organic carbon. The structure of biochar in the form of grits had a size of 5-10 mm. It was produced in the Polish company Fluid.

2. Biochar, with the composition as above, at a dose of 1.6 kg/tree used together with microorganisms of the following composition: a strain belonging to the species *Pseudomonas fluorescens* (Ps1/2) and a strain belonging to the genus *Pantoea* (N52AD), as well as arbuscular mycorrhizal fungi applied in compost at a dose of 0.3 kg compost/tree (375 kg/ha). The mycorrhizal substrate contained spores and hyphae of

arbuscular mycorrhizal fungi: *Glomus caledonium*, *Glomus intraradices* and *Glomus coronatum*.

3. Biochar, with the composition as above, at a dose of 1.6 kg/tree (2000 kg/ha), applied together with Florovit NPK organic fertilizer, at a dose of 0.2 kg/tree (250 kg/ha).
4. Organic fertilizer of the Inco Group, applied at a dose of 0.2 kg/tree (250 kg/ha), and Florovit NPK (N—5%, P—3%, K—2%, organic substance—30%).
5. Microorganisms—bacterial strains belonging to the species *Pseudomonas fluorescens* (Ps1/2) and to the genus *Pantoea* (N52AD). A single application was used in the form of an aqueous suspension in the amount of 200 mL of each strain suspension. The concentration of bacteria in the suspension was $2 \times 10^9$ CFU $\times$ mL$^{-1}$ for the strain Ps1/2 and $1.5 \times 10^9$ CFU $\times$ mL$^{-1}$ for the strain N52AD. Arbuscular mycorrhizal fungi were used in compost (at a dose of 0.3 kg of compost/tree, with the composition presented in Table 2). The mycorrhizal substrate contained spores and hyphae of arbuscular mycorrhizal fungi: *Glomus caledonium*, *Glomus intraradices* and *Glomus coronatum*. The compost was produced at the Institute of Horticulture and was microbiologically enriched.
6. Microorganisms—a strain belonging to the species *Pseudomonas fluorescens* (Ps1/2) and a strain belonging to the genus *Pantoea* (N52AD), and arbuscular mycorrhizal fungi (*Glomus caledonium, Glomus intraradices, Glomus coronatum*), used in compost, with the composition presented in Table 2, were applied at a dose of 0.3 kg of compost/tree (375 kg/ha) together with Florovit NPK organic fertilizer, at a dose of 0.2 kg/tree (250 kg/ha).
7. Control without fertilization

**Table 1.** pH and mineral content of the biochar used.

| pH | P | K | Mg | B | Cu | Fe | Mn | Na | Zn | N og. | C | Organic Matter |
|---|---|---|---|---|---|---|---|---|---|---|---|---|
| KCl | | mg/100 g | | | | [mg kg$^{-1}$] | | | | | % | |
| 6.05 | 85.7 | 58.3 | 22.9 | 14.9 | 6.19 | 219 | 97.2 | 76.3 | 81.3 | 0.96 | 75.9 | 100 |

**Table 2.** pH and mineral content of the compost used.

| pH | P | K | Mg | B | Cu | Fe | Mn | Na | Zn | N og. | C | Organic Matter |
|---|---|---|---|---|---|---|---|---|---|---|---|---|
| KCl | | mg/100 g | | | | [mg kg$^{-1}$] | | | | | % | |
| 6.37 | 10.3 | 21.5 | 11.9 | 1.93 | 3.13 | 1064 | 53.9 | 57.5 | 6.51 | 0.18 | 2.05 | 3.5 |

The biochar, organic fertilization and microbiologically enriched compost were applied in a single dose in May 2014, by sprinkling around tree trunks (in the form of a ring, 0.5 m in diameter), and then mixed with the top layer of soil (up to the depth of 20 cm). Organic fertilization and the application of microorganisms were repeated in the spring of 2015, and then continued in the subsequent years of the experiment. After planting, the trees were trimmed for better rooting and proper growth. In the following years, work was carried out to shorten and bend the shoots, aimed at the correct formation of the tree crown. The trees were watered by drip irrigation. Plant protection was carried out in accordance with the then current recommendations for commercial peach orchards.

Peach leaves were collected for analysis in late July/early August, in the three consecutive years 2015, 2016 and 2017. The determination of the concentrations of minerals in the leaves was performed on a sample of 30 leaves from each tree, picked from the middle part of annual shoots. The dried plant material was mineralized in a mixture of the following concentrated acids: nitric, perchloric and sulphuric, in a microwave oven (Model Ethos-1 from Milistone). The total nitrogen in the plant material was determined by the Dumas method (1826), using the Gerhardt Vapodest apparatus [16]. After mineralization,

the solution was analyzed for macroelements (P, K, Mg, Ca) and microelements (Fe, Mn, Cu, Zn, B) using a sequential plasma spectrometer (Opima 200DV, Perkin-Elmer, Boss and Fredeen, 1999).

The obtained results were statistically analyzed with Statictica 10. One-way analysis of variance was carried out using the Tukey test at a significance level of $\alpha = 0.05$. The results within the same column not significantly different from each other were marked with the same letters.

## 3. Results

The nitrogen content in the leaves fluctuated across the years of the experiment. In 2015, the leaves of trees treated with biochar and organic fertilization had the lowest nitrogen content (2.42) (Table 3). However, in 2016 and 2017, N content in this combination increased by 19% and 34% compared to the year 2015. In 2017, the most nitrogen in peach leaves was found in the combinations where biochar was used together with the organic fertilizer, and microorganisms together with the organic fertilizer, and the least in the treatment with biochar combined with microorganisms. The average results for the three years of research show an increase in leaf nitrogen content after applying biochar and the organic fertilizer (2.86), compared to the control (2.81). The average values for 2015–2017 indicate that the accumulation of phosphorus in peach leaves was best promoted by the application of the organic fertilizer, microorganisms and microorganisms together with the organic fertilizer (Table 3).

**Table 3.** Concentrations of N and P [g 100 g$^{-1}$ DW] in peach leaves in three consecutive years after applying biochar, compost enriched with microorganisms and organic fertilization.

| Treatment | Nitrogen (N) | | | | Phosphorus (P) | | | |
|---|---|---|---|---|---|---|---|---|
| | [g 100 g$^{-1}$ DW] | | | | | | | |
| | 2015 | 2016 | 2017 | 3-Year Average 2015–2017 | 2015 | 2016 | 2017 | 3-Year Average 2015–2017 |
| Biochar | 2.93 ± 0.02 e * | 3.00 ± 0.09 de | 3.02 ± 0.02 e | 2.98 d | 0.26 ± 0.01 b | 0.30 ± 0.01 b | 0.27 ± 0.01 b | 0.27 b |
| Biochar + Microorganisms | 2.78 ± 0.03 d | 2.79 ± 0.03 a | 2.78 ± 0.02 a | 2.78 a | 0.27 ± 0.01 b | 0.30 ± 0.01 b | 0.26 ± 0.02 b | 0.27 b |
| Biochar + Organic fertilizer | 2.42 ± 0.06 a | 2,90 ± 0.06 c | 3.26 ± 0.02 g | 2.86 c | 0.29 ± 0.02 bc | 0.33 ± 0.02 c | 0.31 ± 0.01 c | 0.31 c |
| Organic fertilizer | 2.65 ± 0.9 bc | 3.03 ± 0.07 e | 2.98 ± 0.02 d | 2.89 c | 0.34 ± 0.01 d | 0.35 ± 0.01 d | 0.33 ± 0.01 de | 0.34 d |
| Microorganisms | 2.69 ± 0.15 c | 2.87 ± 0.02 bc | 2.90 ± 0.02 c | 2.82 b | 0.35 ± 0.02 d | 0.32 ± 0.01 c | 0.34 ± 0.01 e | 0.33 cd |
| Microorganisms + Organic fertilizer | 2.63 ± 0.11 b | 2.97 ± 0.03 d | 3.08 ± 0.02 f | 2.89 c | 0.31 ± 0.01 c | 0.36 ± 0.03 d | 0.32 ± 0.01 cd | 0.33 cd |
| Control (without fertilization) | 2.79 ± 0.11 d | 2.83 ± 0.04 ab | 2.82 ± 0.02 b | 2.81 a | 0.23 ± 0.02 a | 0.23 ± 0.01 a | 0.23 ± 0.01 a | 0.23 a |

* Means ± standard error in the same column followed by the same letters do not differ at $p = 0.05$.

In each year of the study, the lowest potassium content (1.78) was found in the leaves of the control plants (Table 4). All the combinations of soil fertilization increased the potassium level in peach leaves relative to the control (without fertilization). However, the best effect was obtained after the application of biochar in combination with organic fertilization, where the average K content in the leaves was 38% higher than in the control leaves. The magnesium content in the leaves of the peach trees fertilized only with biochar was the lowest in each year of the study. The highest Mg content was obtained after using biochar together with the organic fertilizer. The combined use of biochar with organic fertilization increased the magnesium content in the leaves by 20% compared to trees treated with biochar.

**Table 4.** Concentrations of K and Mg [g 100 g$^{-1}$ DW] in peach leaves in three consecutive years after applying biochar, compost enriched with microorganisms and organic fertilization.

| Treatment | Potassium (K) | | | | Magnesium (Mg) | | | |
|---|---|---|---|---|---|---|---|---|
| | [g 100 g$^{-1}$ DW] | | | | | | | |
| | 2015 | 2016 | 2017 | 3-Year Average 2015–2017 | 2015 | 2016 | 2017 | 3-Year Average 2015–2017 |
| Biochar | 2.10 ± 0.02 c * | 2.23 ± 0.02 d | 2.18 ± 0.02 c | 2.17 c | 0.50 ± 0.03 a | 0.48 ± 0.03 a | 0.48 ± 0.03 a | 0.49 a |
| Biochar + Microorganisms | 2.16 ± 0.04 d | 2.28 ± 0.02 e | 2.25 ± 0.02 d | 2.23 c | 0.54 ± 0.03 b | 0.54 ± 0.03 bc | 0.54 ± 0.03 bc | 0.54 b |
| Biochar + Organic fertilizer | 2.42 ± 0.02 f | 2.52 ± 0.02 f | 2.46 ± 0.02 f | 2.47 d | 0.55 ± 0.03 b | 0.63 ± 0.03 e | 0.60 ± 0.03 e | 0.59 c |
| Organic fertilizer | 2.22 ± 0.02 e | 2.18 ± 0.02 c | 2.20 ± 0.02 c | 2.20 c | 0.54 ± 0.03 b | 0.55 ± 0.03 c | 0.55 ± 0.02 c | 0.55 b |
| Microorganisms | 2.04 ± 0.02 b | 2.00 ± 0.02 b | 2.00 ± 0.02 b | 2.01 b | 0.54 ± 0.03 b | 0.53 ± 0.03 b | 0.57 ± 0.03 d | 0.55 b |
| Microorganisms + Organic fertilizer | 2.19 ± 0.06 de | 2.20 ± 0.02 cd | 2.21 ± 0.02 c | 2.20 c | 0.59 ± 0.03 c | 0.57 ± 0.03 d | 0.54 ± 0.03 bc | 0.57 bc |
| Control (without fertilization) | 1.74 ± 0.02 a | 1.77 ± 0.02 a | 1.85 ± 0.02 a | 1.78 a | 0.54 ± 0.03 b | 0.55 ± 0.03 c | 0.53 ± 0.03 b | 0.54 b |

\* Means ± standard error in the same column followed by the same letters do not differ at $p = 0.05$.

The trees fertilized with the organic fertilizer had the lowest calcium content in leaves (2.08). In contrast, the leaves of the trees fertilized with biochar had the highest Ca content (2.41). The results show that the greatest increase in the concentration of boron in the leaves was caused by fertilization with biochar together with the organic fertilizer (26.4), whereas the use of biochar alone did not increase the boron content in the leaves (22.7), relative to the control combination (22.3) (Table 5).

**Table 5.** Concentrations of Ca [mg 100 g$^{-1}$ DW] and B [mg kg$^{-1}$ DW] in peach leaves in three consecutive years after applying biochar, compost enriched with microorganisms and organic fertilization.

| Treatment | Calcium (Ca) | | | | Boron (B) | | | |
|---|---|---|---|---|---|---|---|---|
| | [g 100 g$^{-1}$ DW] | | | | [mg kg$^{-1}$ DW] | | | |
| | 2015 | 2016 | 2017 | 3-Year Average 2015–2017 | 2015 | 2016 | 2017 | 3-Year Average 2015–2017 |
| Biochar | 2.33 ± 0.02 d * | 2.56 ± 0.01 e | 2.34 ± 0.01 e | 2.41 d | 20.6 ± 0.2 a | 22.5 ± 0.4 b | 25.0 ± 0.3 a | 22.7 a |
| Biochar + Microorganisms | 2.23 ± 0.01 c | 2.44 ± 0.01 d | 2.04 ± 0.01 b | 2.24 bc | 20.5 ± 0.3 a | 25.9 ± 0.4 e | 29.3 ± 0.3 e | 25.2 bc |
| Biochar + Organic fertilizer | 2.09 ± 0.01 a | 2.35 ± 0.01 c | 2.46 ± 0.01 f | 2.30 c | 21.3 ± 0.3 b | 26.1 ± 0.3 e | 31.7 ± 0.5 f | 26.4 d |
| Organic fertilizer | 2.07 ± 0.01 a | 2.28 ± 0.01 b | 1.90 ± 0.01 a | 2.08 a | 21.9 ± 0.4 c | 25.2 ± 0.2 d | 28.2 ± 0.2 d | 25.1 bc |
| Microorganisms | 2.22 ± 0.01 c | 2.46 ± 0.02 d | 2.12 ± 0.01 c | 2.27 c | 23.5 ± 0.3 d | 27.3 ± 0.3 f | 26.4 ± 0.2 b | 25.7 c |
| Microorganisms + Organic fertilizer | 2.07 ± 0.01 a | 2.33 ± 0.01 c | 2.18 ± 0.01 d | 2.19 b | 20.7 ± 0.2 a | 24.1 ± 0.1 c | 27.6 ± 0.2 c | 24.1 b |
| Control (without fertilization) | 2.17 ± 0.01 b | 2.10 ± 0.01 a | 2.12 ± 0.01c | 2.13 ab | 20.5 ± 0.2 a | 21.4 ± 0.2 a | 25.1 ± 0.1 a | 22.3 a |

\* Means ± standard error in the same column followed by the same letters do not differ at $p = 0.05$.

The copper content in peach leaves, regardless of the type of treatment, varied from year to year and ranged from 4.08 to 7.74 mg/kg dry weight (Table 6). Both the lowest and the highest Cu content was found in the leaves of the same treatment—with the use of biochar combined with organic fertilization, but in two different years, 2015 (4.08) and 2017 (7.74). Considering the average results from the three years of the study, it should be stated that the highest concentration of copper (6.27) was found in the leaves after the application of microorganisms together with the organic fertilizer, and the lowest in the combinations with microorganisms (5.17) and after the application of biochar (5.13).

**Table 6.** Concentrations of Cu and Fe [mg kg$^{-1}$ DW] in peach leaves in three consecutive years after applying biochar, compost enriched with microorganisms and organic fertilization.

| Treatment | Copper (Cu) | | | | Iron (Fe) | | | |
|---|---|---|---|---|---|---|---|---|
| | [mg kg$^{-1}$ DW] | | | | | | | |
| | 2015 | 2016 | 2017 | 3-Year Average 2015–2017 | 2015 | 2016 | 2017 | 3-Year Average 2015–2017 |
| Biochar | 4.59 ± 0.02 b * | 4.87 ± 0.02 a | 5.94 ± 0.02 c | 5.13 a | 67.0 ± 0.5 b | 63.3 ± 0.3 a | 69.3 ± 0.3 b | 66.5 a |
| Biochar + Microorganisms | 4.70 ± 0.02 bc | 4.89 ± 0.02 a | 6.63 ± 0.02 d | 5.41 c | 78.8 ± 0.4 g | 74.1 ± 0.2 e | 74.7 ± 0.3 d | 75.9 c |
| Biochar + Organic fertilizer | 4.08 ± 0.02 a | 5.87 ± 0.02 d | 7.74 ± 0.02 e | 5.90 d | 62.8 ± 0.4 a | 74.1 ± 0.2 e | 77.3 ± 0.3 f | 71.4 b |
| Organic fertilizer | 4.52 ± 0.02 b | 5.60 ± 0.02 b | 5.71 ± 0.02 b | 5.28 b | 74.4 ± 0.4 f | 72.6 ± 0.3 d | 76.0 ± 0.3 e | 74.3 c |
| Microorganisms | 4.50 ± 0.03 b | 5.72 ± 0.02 c | 5.28 ± 0.02 a | 5.17 a | 70.0 ± 0.5 d | 69.0 ± 0.3 c | 70.6 ± 0.3 c | 69.9 ab |
| Microorganisms + Organic fertilizer | 4.78 ± 0.02 c | 6.33 ± 0.02 e | 7.71 ± 0.03 e | 6.27 e | 68.4 ± 0.4 c | 67.2 ± 0.2 b | 67.9 ± 0.4 a | 67.8 a |
| Control (without fertilization) | 4.82 ± 0.02 c | 5.69 ± 0.02 c | 5.91 ± 0.02 c | 5.47 c | 71.6 ± 0.6 e | 72.8 ± 0.3 d | 77.8 ± 0.3 f | 74.1 c |

* Means ± standard error in the same column followed by the same letters do not differ at $p = 0.05$.

Where only biochar was used, Cu content was lower by 15% relative to its concentration after using biochar combined with organic fertilization. The results from 2015–2017 show a lower iron content in the leaves after the application of microorganisms (69.9), microorganisms with the organic fertilizer (67.8) and biochar (66.5), compared to the control combination (74.1). The highest Fe content (75.9), but not significantly different from the control (74.1), was found in the leaves of the trees fertilized with biochar together with microorganisms (Table 6).

The concentrations of manganese and zinc in peach leaves were the lowest in the combination with microorganisms (49.8 and 12.9) in each year of the study. The highest Mn content was found in the leaves of the trees fertilized with biochar combined with microorganisms (66.6), while the most Zn was found in the leaves of trees fertilized with biochar alone and with the organic fertilizer alone (17.7 and 17.4) (Table 7). The combined application of biochar and organic fertilization slightly increased the concentrations of manganese and zinc in peach leaves, but in the case of these elements, the leaves had a higher Mn and Zn content after using biochar only.

**Table 7.** Concentrations of Mn and Zn [mg kg$^{-1}$ DW] in peach leaves in three consecutive years after applying biochar, compost enriched with microorganisms and organic fertilization.

| Treatment | Manganese (Mn) | | | | Zinc (Zn) | | | |
|---|---|---|---|---|---|---|---|---|
| | [mg kg$^{-1}$ DW] | | | | | | | |
| | 2015 | 2016 | 2017 | 3-Year Average 2015–2017 | 2015 | 2016 | 2017 | 3-Year Average 2015–2017 |
| Biochar | 61.5 ± 0.2 e * | 62.7 ± 0.3 e | 61.9 ± 0.4 e | 62.0 c | 16.1 ± 0.2 d | 18.3 ± 0.3 d | 18.7 ± 0.2 e | 17.7 c |
| Biochar + Microorganisms | 64.7 ± 0.3 f | 66.5 ± 0.4 f | 68.7 ± 0.3 f | 66.6 d | 14.8 ± 0.3 c | 16.0 ± 0.2 c | 18.7 ± 0.3 e | 16.5 bc |
| Biochar + Organic fertilizer | 54.4 ± 0.3 c | 56.6 ± 0.3 c | 57.0 ± 0.3 d | 56.0 b | 15.1 ± 0.3 c | 15.4 ± 0.2 b | 17.1 ± 0.4 bc | 15.9 bc |
| Organic fertilizer | 54.9 ± 0.4 cd | 57.3 ± 0.3 d | 55.9 ± 0.4 c | 56.0 b | 16.5 ± 0.2 d | 17.9 ± 0.2 d | 17.8 ± 0.2 d | 17.4 bc |
| Microorganisms | 48.9 ± 0.4 a | 49.8 ± 0.2 a | 50.7 ± 0.3 a | 49.8 a | 11.8 ± 0.2 a | 13.3 ± 0.3 a | 13.6 ± 0.2 a | 12.9 a |
| Microorganisms + Organic fertilizer | 52.6 ± 0.3 b | 54.1 ± 0.3 b | 54.2 ± 0.3 b | 53.6 b | 13.5 ± 0.1 b | 15.7 ± 0.2 bc | 17.4 ± 0.2 cd | 15.5 b |
| Control (without fertilization) | 55.5 ± 0.5 d | 56.2 ± 0.2 c | 55.6 ± 0.3 a | 55.8 b | 13.8 ± 0.3 b | 15.7 ± 0.2 bc | 16.8 ± 0.2 b | 15.4 b |

* Means ± standard error in the same column followed by the same letters do not differ at $p = 0.05$.

## 4. Discussion

The use of biochar in the cultivation of fruit trees has a short history, and therefore there is a little information on this subject in the literature. The use of a small dose of biochar (2000 kg/ha) in our research has shown that it is an effective and economically justified dose. This is especially with the combination of a low dose of biochar with organic fertilization, which gives very good results in the fertilization of peach trees. It is best seen in the example of nitrogen.

The results of our own research are consistent with the results obtained by Adekiya et al. [12]. In the experiment of these authors, the application of biochar alone did not increase the N content, while the combined application of fertilizer with biochar significantly increased nitrogen content in the leaves of radish.

Dharmakeerthi et al. [17] confirmed the lack of an effect of biochar alone on increasing the concentration of N in leaves too. The combined use of biochar and mineral fertilization also did not increase the nitrogen content in the leaves, but the quality of the trees that produced *Hevea brasiliensis* was higher.

Milošević et al. [18], comparing the effects of organic and inorganic fertilization, showed that the applied manure had a greater effect on the nitrogen content in apricot leaves than mineral NPK fertilization. In contrast, the concentrations of P and K in the leaves of the plants treated with the mineral fertilizer were higher than in the case of fertilization with manure. In our study, we observed, over the three consecutive years, an increasing nitrogen content in peach leaves as a result of applying biochar in combination with organic fertilization, but the differences were not statistically significant. However, the doses of biochar and organic fertilizer were lower than in the experiment by Adekiya et al. [12]. The use of biochar as reported by Gunes et al. [19] increased the concentrations of nitrogen, phosphorus and potassium in lettuce leaves, and at the same time decreased the content of iron, copper, zinc and manganese. In the third year of our own research, the highest content of copper, zinc, manganese, calcium and boron was observed in tree leaves in combinations where biochar was used. Gunes et al. [20], using pyrolysed (at four temperatures) biochar, found that the concentrations of phosphorus and potassium in lettuce were the highest after applying to the soil biochar obtained at temperatures of 300 and 350 °C. The concentrations of these elements were higher in the control plants. Studies of some authors have indicated that the application of biochar to calcareous and sandy soils significantly increased the concentrations of nitrogen and phosphorus in zucchini fruit and plants, and improved soil fertility [14]. The results obtained by Saletnik et al. [15] showed a positive effect of biochar on the potassium content in plants, and its concentration increased with the increase in the dose of biochar (10, 15, 20 g) applied to the soil. Moreover, the combined use of biochar with ash significantly increased the potassium content in the leaves of the tested plants. Significant differences in the concentrations of phosphorus and calcium in plants, compared to the control plants, were also evident after the combined application of biochar and ash. Wacal et al. [21] noticed that biochar in a dose of 50 t ha$^{-1}$ had the greatest impact on the potassium (K) content in sesame leaves at the first harvest. It also modified the nitrogen content in the leaves to a lesser extent. However, the chemical analysis of the second harvest of sesame did not show a significant increase in nitrogen, potassium, phosphorus, calcium or magnesium. In our study, the concentrations of nitrogen, phosphorus and potassium in peach leaves varied markedly, both under the influence of the applied biochar and organic fertilization. In the case of iron (Fe), the leaves of peach trees fertilized with biochar had a lower concentration of this element (66.5) than the leaves of control trees (74.1). Gale and Thomas [22], using different doses of biochar in the cultivation of velvetleaf (*Abutilon theophrasti*), stated that after using biochar in doses greater than 10 t ha$^{-1}$, the nitrogen (N) content in the leaves decreased significantly (by approx. 30%). On the other hand, the boron content in the leaves after applying biochar at 10 t ha$^{-1}$ increased twofold. In our study, the boron content increased in the subsequent years of the experiment, to the greatest extent after the combined application of biochar and the organic fertilizer (from 21.3 to 31.7); these differences were statistically significant.

Research by Adekiya et al. [12] confirms the influence of the combined application of biochar with organic fertilization on increasing the phosphorus content in radish leaves. As reported by Amin and Eissa [14], the use of biochar (6.3 g per pot) increased the concentration of nitrogen (N) in zucchini fruit by 39% relative to the control but increasing the dose of biochar to 12.6 and 26.5 g did not cause greater accumulation of nitrogen in the fruit. Irrespective of the dose, biochar did not affect nitrogen accumulation in the plants. Neither did biochar increase the phosphorus (P) content in zucchini fruit, but when applied at a dose of 12.6 g per pot it increased the phosphorus content in plants by 55% relative to the control. In our research, the use of biochar did not increase the phosphorus content in peach leaves. However, organic fertilizer increased the content of this element. A higher P concentration was recorded in all combinations where organic fertilization was applied. Amin [23], analyzing the effects of three doses of biochar (20, 40 and 60 t ha$^{-1}$), noted that with an increase in the dose of biochar to the soil, the fresh and dry weight of wheat plants increased and the concentration of K in the tested plants also increased (1.24–1.49), when compared to the control plants (1.14), and the differences were statistically significant. In an experiment conducted in the Experimental Orchard in Dąbrowice, we found the highest level of K in the leaves of those peach trees where biochar was applied in combination with organic fertilization (2.47). The combined use of biochar and organic fertilization significantly increased the yield of pear fruit and the estimated leaf chlorophyll content (SPAD) relative to the control (Schaffert and Percival [24]). Increased chlorophyll content in leaves (SPAD), resulting from the use of biochar in wheat cultivation, was also observed by Salim [25]; the application of 2% biochar to the soil increased the degree of leaf colouring by 24%. The addition of 5% biochar to a sandy soil increased the colour of the leaves by 17%, compared to the control plants. Earlier, our own research conducted on peach trees showed [26] the beneficial effect of the combined use of biochar and organic fertilization on the leaf size as well as the diameter of the trunk. In addition, in the third year after application, a significant increase in yield and fruit quantity was observed. The proposed treatments may be useful in fruit production using organic methods.

Similar conclusions were reached by Steiner et al. [27], who showed better efficiency in plant growth and yield of fertilizers used together with biochar, compared to fertilization alone in rice and sorghum cultivation.

According to Wójcik [28], the optimum content of ingredients in peach leaves in the soil and climatic conditions of Poland is N 2.31–4.00%, P 0.14–0.30%, K 1.91–3.00%, Mg 0.30–0.60%, B 20–40 mg/kg$^{-1}$, Cu 5–10 mg/kg$^{-1}$, Mn 50–100 mg/kg$^{-1}$, Zn 20–40 mg/kg$^{-1}$. There is no literature data on the optimal content of calcium and iron in peach leaves. Regardless of the experimental combinations used, the content of N, P, K, Mg, B and Mn in the leaves was within the optimal range or was close to it. Lower values were found for copper and zinc. The uptake of these microelements from the soil is most intense at pH 5–5.6, and in our experiment the soil pH was 6.2.

## 5. Conclusions

The significant increase in nitrogen content was observed in each subsequent year in the leaves of peach trees after the combined application of biochar and organic fertilization. The increased concentrations of potassium, magnesium, calcium, boron and copper in the leaves from one year to the next clearly indicate the positive influence of the combined use of biochar with organic fertilization. The average concentrations of copper, iron, boron, potassium, magnesium and phosphorus in peach leaves were significantly lower where only biochar was used, compared with the combined application of biochar and organic fertilization. The effect of applying biochar to the soil in combination with organic fertilization was most evident in the third year of the study.

In general, the conducted experiment shows that the combined use of biochar and organic fertilizer is visible over a longer period of time after their application. More research is needed on the use of microorganisms and methods of their application in the orchards and on various species of fruit trees.

**Author Contributions:** Conceptualization, M.F. and L.S.-P. and M.S.; methodology, M.F. and L.S.-P. and M.S.; investigation, M.F.; writing—original draft, M.S.; writing—review and editing, L.S.-P.; funding acquisition, L.S.-P. All authors have read and agreed to the published version of the manuscript.

**Funding:** This research received no external funding.

**Institutional Review Board Statement:** Not applicable.

**Data Availability Statement:** The data presented in this study are available on request from the corresponding author.

**Conflicts of Interest:** The authors declare no conflict of interest.

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
