# Peer review of "Macro and Microelements in Leaves of ‘Meredith’ Peach Cultivar Supplied with Biochar, Organic and Beneficial Biofertilizer Combinations"

_agriculture, doi:10.3390/agriculture13050933_

Round 1

Reviewer 1 Report (New Reviewer)

Review Report

Title: Content of macro and microelements in the leaves after soil application of biochar, organic fertilization and beneficial microorganisms

Modified the title Influence of soil application of biochar, organic fertilization and microorganisms improved the mineral nutrients in the leaves of ‘Mere- 2 dith’ peach cultivar

In this paper, the authors investigated the role of biochar, organic fertilization and microorganisms to improve the mineral contents in the leaves of Peach plant. The work has novel information to understand the mechanism of organic amendment to enhance the minerals nutrients in peach leaves. Although article can be interesting for the readers of the agriculture. Nevertheless, the language of the article is poor, along with long and ambiguous sentences. I am also not convinced with the outcomes of this study, and the conclusion authors actually had made.

Comments to authors

1. In the abstract, please mention treatments of study are not presented in detail; especially concentration of biochar; levels of organic fertilization and also sources; and mention species of micro-organism in the abstract

3. Add the solid conclusion at the end of the abstract section according to study hypothesis
4. Add the results in the abstract in percentile like how much increase through various treatment or how much percent decrease. 

6. Hypothesis of the study is missing; add the study gap (what is new in this study?). Authors should provide a clear research question and explain clearly what is new about your work and provide a clear hypothesis.

7. In materials and methods, experiment design is missing

8. Why author used only one concentration of biochar ----- applied at a dose of 1.6 kg/tree (2000 kg/ha). Any reference or study before on this?

8. What was age of peach plant, when treatments were applied? Treatments were applied every three years or at one time at transplanting   

9. Which leaves of peach plants were used for minerals analysis old or new leaves?

8. Please use the Standard error with mean Means ± SE

10. Results and discussion section of the article is weak, authors mainly focused on their results but they did not discuss them according to international standards. Moreover the writing style of results and discussion section is also ambiguous, with long and weak sentences and in a repetitive way. I am not convinced with the way of discussion of the authors, in its current form it cannot be accepted in agriculture. I will recommend a thorough revision of this section.

12. The conclusions is too much long, please shortened and should answer the hypothesis of your study and should focus on the implication of your findings. Please, avoid using abbreviations and acronyms in this section

13. Language, wording and paraphrasing should be carefully reviewed and improved. A native English-speaking scientist or professional English editing service must edit your manuscript.
14. The reference of the article needs to be checked, revised and formatted.

Author Response

Reviewer 2 Report (New Reviewer)

The paper by Frac et al on the effect of biochar and fertiliser on foliar nutrients of peach trees is overall well written. There are some major issues with the presentation/discussion of the results that must be addressed. Currently, the discussion starts with published data, rather than the results obtained by the researchers. This needs a complete re-write, starting with your results, how these compare to other work, explain your results and then arrive at a conclusion. At this stage, your paper explains other people's data not your own. Furthermore, you paper is purely descriptive, it does not offer a mechanistic understanding of the processes giving rise to the data. This is a shortcoming that needs to be addressed.

Other minor issues which need to be addressed:

Are these trees grafted on clonal root stocks, or are these bare-rooted plants? Rootstocks significantly affect nutrient uptake in trees, if these were grafted on non-clonal rootstocks, the results are virtually meaningless.

The low rate of nutrient application, while realistic, will always make it difficult to see treatment differences. I do not suggest that higher rates of biochar or compost should have been used - numerous papers use unrealistically high application rates and I commen the authors for using lower rates. But it should be mentioned in their discussion.

Please also express fertiliser composition in terms of % element, not % oxide.

Can you include a statement showing the adequate range of foliar nutrients in this peach variety (or other peach varieties) - were the plant nutrients in the adequate range?

Considering these are young trees, I am not convinced that the nutrient application rates are sufficient to meet nutrient replacement rates in fruit-bearing trees. Can you comment on this? The mineral fertiliser was applied at rates supplying 12 kg N/ha and 3 kg P/ha. What is the purpose of such low rates?

In the first year data, the nitrogen content decreased in the compost and biochar treatments. This suggests to me there was nitrogen drawdown in these treatments. Do you have the C/N ratios for the amendments?

Author Response

Reviewer 3 Report (New Reviewer)

Undoubtedly, the authors have done a great contribution to research of macro and microelements of peach leaves after soil application of biochar, organic fertilization and beneficial microorganisms. The manuscript can be recommended for publication after the elimination of the following remarks:

Lines 107-108 “The aim of the research carried out in this study was to determine the effects of bio-char and organic fertilization on the concentrations of macro- and microelements in the leaves of the peach cultivar ‘Meredith’ ”.  The term 'research' is broader than the term 'study'. Thus, you should swap them.

Line 112 – Specify the age of the trees.

Table 3, lines 263-267, 334-336 Judging by the experimental data, when studying the effect of biochar and its combinations with microorganisms and organic fertilizers on the nitrogen content, you did not take into account other factors, so the distribution of N concentrations does not follow a regularity over the years.

Table 3-7 The results of the study by years can be presented in additional materials to the manuscript, and only averaged data can be left in the text, which will significantly reduce the volume of the text and simplify the presentation of the results. It is recommended to combine the tables into 2 general ones - macroelements (P, K, Mg, Ca) and microelements (Fe, Mn, Cu, Zn, B).

Lines 267-288 This text does not contain analysis and comparison with your research. It should either be supplemented with comparisons, or this passage should be moved to the Introduction section.

Compare your results for the content of elements in peach leaves with the optimal ones and make a conclusion about whether the plants received enough nutrition using the techniques you developed.

Line 301 You first state that “the use of biochar (6.3 g per pot) increased the concentration of nitrogen in zucchini fruit by 39%”, then “…biochar did not affect nitrogen accumulation in the plants” (line 304). Please check and give reference or change the text.

Lines 323-328 As I understand it, this is the same study that you describe in this manuscript. In my opinion, these data should not be shared. The action of biochar did not have a sustainable effect over the years of research: “In the first growing season, no positive changes were found after the use of biochar” [26]. How can you explain this effect of biochar? Can its digestibility depend on weather conditions (temperature, rainfall, etc.) or on accumulation in the soil?

Lines 334-350 I think it would be better to rewrite the Conclusions section, outlining the best treatments and giving recommendations on how to use the fertilizers.

Lines 352-422 Please complete the list of references according to the requirements of the journal.

In general, the text lacks a chemical justification for the consumption of ions (the antagonism and synergism of macro- and microelements is not taken into account), the final goal of the study is not physiologically justified, there is no clear picture of what for the treatments were ultimately carried out and how it affected the final yield (compare with [26]). Please, add additional information to the Discussion section as recommended.

Round 2

Reviewer 1 Report (New Reviewer)

I think the author has revised as requested, I have no further comments.

Author Response

Thank you for your review

Reviewer 3 Report (New Reviewer)

Authors have done a great work of correcting the manuscript. However, I still have notes for correction.

I recommend you to reduce the Abstract. You can introduce abbreviations for experience options.

Line 126 “An attempt was made...” It’s not the best turn of phrase for a scientific article. Rephrase.

Line 170 Taking into account that you are offering your biochar treatments and organic fertilization as a replacement for traditional mineral fertilization, it was reasonable to use the recommended mineral fertilization as a control for comparison. It is obvious that without fertilizers, high yields and accumulation of nutrients will not be observed at all. How can we evaluate the benefits of your treatments over traditional ones? If you have not done such an analysis, I recommend at least adding a comparison of the elemental composition to the literature review. A comparative assessment is needed for practical horticulture in order to assess possible losses and risks.

I also recommend that you mention that the treatments you propose will be useful for obtaining organic products.

Line 124-126 “The aim of the study was to determine the effects of biochar, organic fertilization and beneficial microorganisms on the concentrations of macro- and microelements in the leaves of ‘Meredith’ peach trees”. Please complete the aim. What is this for? Maybe it's for organic farming or to reduce fertilizer costs? Reflect the practical significance of the work in the purpose of the study.

You have greatly improved the discussion section, however I would like you to add some information on how increasing or decreasing elements per experience can affect, for example, flower bud set or yield. This can be specified for the best variant of the experiment and/or the lowest values of element concentrations.

Correct in text: “Bacteria” is a plural word; the singular is bacterium. Change “bacterias” in the text.

Author Response

This manuscript is a resubmission of an earlier submission. The following is a list of the peer review reports and author responses from that submission.

Round 1

Reviewer 1 Report

The article is presented very practically from both a scientific and a practical point of view. It discusses the possibilities of how to improve the supply of micro and macro elements using bio fertilizers and their modifications, which is not only friendly to the environment and consumers, but also more natural for plants.The effect of applying biochar to the soil in combination with organic fertilization in peach cultivation was most evident in the third year of the study. This experiment clearly shows that the influence of biochar applied to the soil, and especially its positive effect on plants, is evident in the following years, not necessarily immediately after its application, and that applying biochar in combination with an organic fertilizer gives even better results.

Author Response

Authors would like to thank you for your time and effort to review the manuscript. Thank you for your positive assessment.

Reviewer 2 Report

Dear Editor,

I carefully read the submission title "Influence of biochar, organic fertilization, and beneficial soil microorganisms on the concentrations of macro- and microelements in the leaves of ‘Meredith’ peach trees ".

In fact more recently there has been an increasing interest to biochar and bio fertilizers.  

My first impression that the paper contains new information and title of the manuscript cover its content. The summary is appropriate and the aim of the work clearly established. The methods are used are adequate and used sophisticated techniques and equipment's. I found the results with a valuable data. Discussion and conclusions are well documented.

However, I have some corrections and additions on it before acceptance.

INTRODUCTION: Please add more sentences talking about the importance of soil microorganisms used in this study. Please use more references to make this stronger

I suggest below ones

Jacoby R, Peukert M, Succurro A, Koprivova A, Kopriva S. The Role of Soil Microorganisms in Plant Mineral Nutrition-Current Knowledge and Future Directions. Front Plant Sci. 2017 Sep 19;8:1617. doi: 10.3389/fpls.2017.01617. PMID: 28974956; PMCID: PMC5610682.

Suresh P, Rekha M, Gomathinayagam S, Ramamoorthy V, Sharma MP, Sakthivel P, Sekar K, Valan Arasu M, Shanmugaiah V. Characterization and Assessment of 2, 4-Diacetylphloroglucinol (DAPG)-Producing Pseudomonas fluorescens VSMKU3054 for the Management of Tomato Bacterial Wilt.

M & M section

-          It need to add a section about the effect of different treatment in yield quantity and quality

- Also, study the cytotoxicity of different treatments the concentrations of macromolecules, it many times led to many chromosomal aberrations).

- Physiological and biochemical studies at control and after treatments.  

Results

It needed to be supported with images of plants at control and treatments cases

Reviewer 3 Report

The manuscript describes the effect of biochar, organic fertilization, and beneficial soil microorganisms on the concentrations of macro- and micro-elements in the leaves of ‘Meredith’ peach trees. The author analyzed the data of three years, but lacked statistical analysis to explain the correlation of biochar, organic fertilization, microorganisms and elements of leaves. Readers may not be able to see the meaning and the importance of this manuscript. In particular, the abstract and discussion should be reorganized and rewritten.

1.      What do you want to explain in the abstract? The abstract lacks the conclusion. Only element concentration was described. What was the relationship among biochar, organic fertilization, microorganisms and elements? The author should emphasize the relationship of these, rather than describe the results. The abstract should be rearranged and rewritten.

2.      Line 17, the number of 2.98-2.81….. I don't think these raw data need to appear in the abstract. These data also lack concentration units.

3.      Line 18, The accumulation of phosphorus in peach leaves was best promoted by the use of the organic fertilizer, microorganisms, and microorganisms together with the organic fertilizer. Which one is the best? According to your result, the best promotion was the use of the organic fertilizer.

4.      I wonder why there is no description of potassium in the abstract?

5.      The author should explain what consequence is expected under different fertilization conditions.

6.      What is the hypothesis of the manuscript?

7.      Line 115, What kind of arbuscular mycorrhizal was applied?

8.      Line 129, Glomus intraradices, and Glomus coronatum should be written with full name.

9.      The repeated measures ANOVA could be applied in the analysis of biochar, organic fertilization, and beneficial soil microorganisms across 3 years data.

10.  The RDA analysis also could be applied in the manuscript.

11.  More relationship according to statistics need to be added to the discussion.

12.  In discussion, only the comparison of data with previous studies can not find out the value of the manuscript.

13.  Why only emphasize the analysis of N, P, and K in the discussion? How about Mg, Ca, B, Cu, Fe, Mn, and Zn?

14.  The discussion should be rewritten and the statistical analysis should be added.

15.  The manuscript submitted to the section of The Impact of Environmental Factors on Fruit Quality. However, the manuscript lacks the evaluation of fruit quality. If the data of fruit quality are added, the relationship between fertilization and fruit can be confirmed.

Round 2

Reviewer 2 Report

The authors made the required changes and modifications to improve their manuscript.

Author Response

The authors would like to thank you for the time and effort put into the review, and for the valuable comments improving the value of the publication.

Reviewer 3 Report

Although this manuscript has made the element analysis in the leaves of ‘Meredith’ peach trees, it lacks the discussion of relevance. The author has no point-to-point reply of reviewer's questions such as the question 9, 10, 11 and 15. The author should use statistics for discussion and analysis such as repeated measures ANOVA or RDA analysis.

Author Response

The authors would like to thank you for the time and effort put into the review, and for the valuable comments improving the value of the publication.

For questions 9,10, 11 and 12 was given one answer:
"The results were analyzed using the ANOVA statistical method as suggested by the reviewer, both for individual years and for the three-year mean. The dependencies observed during the three-year study were described in the conclusions."

The answer to question 15 was mistakenly not entered on the reviewer's answer sheet. The reply was sent only to the editor. Text of this reply: "All these comments have been included in the amendments. In the discussion part, a few references to research on the quality of fruit were placed".
